# Talazoparib Does Not Interact with ABCB1 Transporter or Cytochrome P450s, but Modulates Multidrug Resistance Mediated by ABCC1 and ABCG2: An in Vitro and Ex Vivo Study

**DOI:** 10.3390/ijms232214338

**Published:** 2022-11-18

**Authors:** Ziba Sabet, Dimitrios Vagiannis, Youssif Budagaga, Yu Zhang, Eva Novotná, Ivo Hanke, Tomáš Rozkoš, Jakub Hofman

**Affiliations:** 1Department of Pharmacology and Toxicology, Faculty of Pharmacy in Hradec Králové, Charles University, Heyrovskeho 1203, 500 05 Hradec Králové, Czech Republic; 2Department of Biochemical Sciences, Faculty of Pharmacy in Hradec Králové, Charles University, Heyrovskeho 1203, 500 05 Hradec Králové, Czech Republic; 3Department of Cardiac Surgery, Faculty of Medicine, Charles University in Hradec Králové and University Hospital Hradec Králové, Sokolská 581, 500 05 Hradec Králové, Czech Republic; 4The Fingerland Department of Pathology, Charles University, Faculty of Medicine and University Hospital in Hradec Králové, Czech Republic, Sokolská 581, 500 05 Hradec Králové, Czech Republic

**Keywords:** talazoparib, ABC transporter, small cell lung cancer, multidrug resistance, pharmacokinetic drug–drug interaction, cytochrome P450

## Abstract

Talazoparib (Talzenna) is a novel poly (adenosine diphosphate-ribose) polymerase (PARP) inhibitor that is clinically used for the therapy of breast cancer. Furthermore, the drug has shown antitumor activity against different cancer types, including non-small cell lung cancer (NSCLC). In this work, we investigated the possible inhibitory interactions of talazoparib toward selected ATP-binding cassette (ABC) drug efflux transporters and cytochrome P450 biotransformation enzymes (CYPs) and evaluated its position in multidrug resistance (MDR). In accumulation studies, talazoparib interacted with the ABCC1 and ABCG2 transporters, but there were no significant effects on ABCB1. Furthermore, incubation assays revealed a negligible capacity of the tested drug to inhibit clinically relevant CYPs. In in vitro drug combination experiments, talazoparib synergistically reversed daunorubicin and mitoxantrone resistance in cells with ABCC1 and ABCG2 expression, respectively. Importantly, the position of an effective MDR modulator was further confirmed in drug combinations performed in ex vivo NSCLC patients-derived explants, whereas the possible victim role was refuted in comparative proliferation experiments. In addition, talazoparib had no significant effects on the mRNA-level expressions of MDR-related ABC transporters in the MCF-7 cellular model. In summary, our study presents a comprehensive overview on the pharmacokinetic drug–drug interactions (DDI) profile of talazoparib. Moreover, we introduced talazoparib as an efficient MDR antagonist.

## 1. Introduction

Oncological diseases are characterized by specific gene mutations and/or epigenetic dysregulations, which drive the transformation of healthy tissues to malignant tumors. Among all kinds of cancers, lung cancer is one of the most common variants, with the highest mortality (more than 1.38 million annually) in both genders all around the world [1,2]. Within lung cancers, around 85% of tumors are diagnosed as non-small cell lung cancer (NSCLC) [3]. Due to its high prevalence and poor prognosis, significant attention is paid to the development of novel curative approaches. NSCLC is traditionally treated with chemotherapy, radiation, and surgery [4]. In chemotherapy, the drugs target rapidly diving cells with limited discrimination between healthy and malignant cells. This feature leads to the occurrence of harmful adverse effects and substantially overshadow the clinical success of conventional chemotherapies [5]. In contrast, new targeted anticancer therapeutics are designed to interfere with cancer-specific pathways, thus producing high efficiency accompanied by acceptable toxicity [6,7]. Beside other pathways, poly (adenosine diphosphate-ribose) polymerase (PARP) has emerged as a unique druggable target, which literally revolutionized the therapy of ovarian and breast cancers. PARP is a DNA-repair enzyme necessary for the reparation of single-strand breaks in genomic DNA. The blockade of its function follows the concept of synthetic lethality and induces apoptosis selectively in cells with a *BRCA* mutation [8,9]. The clinical success of PARP inhibitors (PARPi) began with the approval of olaparib in 2014; the drugs from this group have shown outstanding improvement of progression-free survival in comparison with conventional chemotherapy. Talazoparib (BMN-673; trade name Talzenna; Figure 1) is a fourth-in-class drug and was approved by the US Food and Drug Administration (US FDA) for the treatment of advanced or metastatic breast cancer patients with a germline *BRCA* mutation in 2018 [10,11]. *BRCA* mutation is frequent in breast and ovarian cancers; however, it is also present in some other cancer types to a lower extent [12,13]. Therefore, talazoparib has been evaluated in clinical trials for the therapy of several hematological and solid malignancies, including NSCLC (e.g., NCT03426254, NCT03377556, NCT04173507, and NCT02693535).

The introduction of pharmacotherapy has helped to transform an anticancer treatment; however, even the most effective drugs face the problem of multidrug resistance (MDR). This major clinical obstacle leading to therapy failure is tightly associated with the intrinsic genetic variability of cancer, which allows for the development of molecular mechanisms evading drugs’ effects [14]. Resistance occurs both in conventional chemotherapeutics as well as novel targeted drugs. MDR is based on the combination of different pharmacokinetic and pharmacodynamic principles, including upregulated drug efflux and metabolism, downregulated drug influx, mutation of drug’s target, and enhancement of DNA repair [15,16]. Within pharmacokinetic aspects, several members of the ATP-binding cassette (ABC) drug transporter superfamily mediate anticancer drug efflux and thus contribute to MDR [17,18]. The essential participation in this process has been described for P-glycoprotein (ABCB1), breast cancer resistance protein (ABCG2), and multidrug resistance-associated protein 1 (ABCC1) [19]. Next to the function in MDR, ABC efflux transporters play an important tissue-protective role and affect systemic drug disposition, while the first two mentioned transporters are recognized as important sites for pharmacokinetic drug–drug interactions (DDIs) [20]. Drug efflux transporters form a cooperative unit with drug-metabolizing enzymes, which also have two opposite faces. Metabolism is an irreplaceable biological process that affects drugs’ pharmacodynamic activities and pharmacokinetic behavior (absorption and elimination), thus assisting with detoxication. At the same time, some enzymes have been found to be overexpressed in tumors, where they force the deactivation of anticancer agents. Recently, we described the role of 3A4 isoform of cytochrome P450 (CYP) in the resistance to docetaxel [21]. CYPs superfamily represents a dominant group of phase-I metabolic enzymes. Similar to ABCB1/ABCG2, several members (CYP1A2, CYP2B6, CYP2C8, CYP2C9, CYP2C19, CYP2D6, CYP3A4, and CYP3A5) are mediators of pharmacokinetic DDIs [22,23]. The knowledge on novel drugs’ interactions with ABC transporters and CYPs is critical for clinicians to avoid the prescription of potentially dangerous combinations [24,25]. At the same time, inhibitory interactions can be beneficially exploited for targeting MDR. While the clinical translation of the idea of nontoxic MDR modulators have failed, the dual-activity chemosensitizers from the group of targeted anticancer agents currently attract considerable attention [18,26,27]. Such drugs simultaneously show their own anticancer effects and inhibit transporter/enzyme-mediated MDR to conventional chemotherapeutics. Their combination thus produces synergistic effects, and due to the targeted character of dual-activity modulators, this approach has potential to overcome the limits of old-fashioned MDR modulation.

In this study, we investigated the interactions of talazoparib with ABC drug efflux transporters and CYPs participating in MDR and/or DDIs. Furthermore, we examined the possible modulator or victim roles of this PARPi in the MDR phenomenon. Appropriate in vitro and ex vivo techniques were used for this purpose, including cell accumulation studies, incubations with recombinant enzymes, drug combination assays, comparative proliferation experiments, and gene induction studies.

## 2. Results

### 2.1. Talazoparib Significantly Inhibits ABCG2- and ABCC1-Mediated Transport of Probe Substrate Drugs

First, we employed accumulation studies in MDCKII-par, MDCKII-ABCB1, MDCKII-ABCC1, and MDCKII-ABCG2 cells with fluorescent MDR-victim anticancer drugs to determine possible transporter-inhibitory effects of talazoparib. In the case of MDCKII-ABCB1, no inhibitory interaction occurred (IC_50_ ˃ 50 µM) (Figure 2A). In contrast, in MDCKII-ABCG2, a modest effect of talazoparib on ABCG2 efflux activity was observed with IC_50_ of 11.2 µM (Figure 2B). Finally, in MDCKII-ABCC1, talazoparib potently inhibited ABCC1-mediated transport of daunorubicin, showing IC_50_ of 5.62 µM (Figure 2C). To validate the data, experiments were also performed in control parent cells. As expected, tested PARPi did not exhibit any significant effects on the accumulations of probe substrate drugs.

### 2.2. Talazoparib Is Predicted to Interact with the Ligand-Binding Sites of ABCG2 and ABCC1 and Nucleotide Binding Domain 2 of ABCC1

Molecular modeling was performed to explain the molecular background of outcomes recorded in the preceding accumulation studies. Flexible molecular docking to ABCG2 predicted the high affinity of talazoparib for the internal cavity (ligand-binding site) of the transporter (−11.9 kcal/mol). As ABCG2 exists as a dimer, talazoparib could potentially bind to both monomers (Figure 3A) interacting with residues Phe-431, Phe-432, Thr-435, Phe-439, Thr-542, and Val-546 (Figure 3B).

For ABCC1, no human crystallographic structures with both ATP/ADP molecules or a ligand are available in the PDB database. Therefore, prior to our calculations, the models were created using bovine ABCC1 crystals and the human ABCC1 primary sequence. As described for leukotriene C4 (LTC4) cocrystallized with the protein backbone (PDB ID: 5UJA) [28], a positively charged P-pocket coordinating the glutathione moiety and a hydrophobic H-pocket surrounding the LTC4 lipid tail can be distinguished at the binding site. Based on our calculations, talazoparib binds to the P-pocket (−11.7 kcal/mol), where it interacts with residues His-335, Phe-385, Tyr-440, Phe-594, Asn-1245, Trp-1246, and Arg-1249 (Figure 4A,B).

Talazoparib, as the potent PARP inhibitor, was originally designed on the basis of a nicotinamide-like pharmacophore. Its pyridazinone moiety mimics NAD^+^, which serves as a substrate for PARP polymerase and provides ADP-ribose monomers for the synthesis of poly-ADP-ribose chains [29]. Therefore, we were interested in the hypothesis that talazoparib might interact with amino acid residues of the nucleotide binding domains (NBDs) of examined ABC transporters. As shown in Figure 4C, it was predicted that talazoparib interacts with NBD2 residues (−10.7 kcal/mol) of ABCC1. In contrast, no conformation with binding affinity less than −10.0 kcal/mol was detected for ABCC1′s NBD1 and both NBDs of the ABCG2 transporter.

### 2.3. Talazoparib Does Not Interfere with the Metabolic Activities of Clinically Relevant CYP Isoforms

In this experimental set, we screened talazoparib’s possible inhibitory effects on clinically relevant human CYP isozymes. We selected eight isoforms playing a role in pharmacokinetic DDIs (CYP1A2, CYP2B6, CYP2C8, CYP2C9, CYP2C19, CYP2D6, CYP3A4, and CYP3A5) [24,25] and docetaxel resistance (CYP3A4) [21]. Talazoparib negligibly inhibited the metabolic activities of all the examined CYPs with IC_50_ ˃ 50 µM (Figure 5). The drug is thus unlikely to stand in the position of the perpetrator of metabolic inhibitory DDIs or modulator of CYP3A4-mediated MDR.

### 2.4. Talazoparib Combats ABCC1- and ABCG2-Mediated Chemotherapeutic Resistance In Vitro

Combination assays were carried out to evaluate talazoparib’s potential to become a dual-activity MDR modulator. Daunorubicin and mitoxantrone were selected as MDR-susceptible substrate drugs for ABCC1- and ABCG2-oriented studies, respectively. The modulatory concentration of talazoparib (5 µM) was chosen, as it induced significant ABCC1 and ABCG2 inhibition and, at the same time, had negligible cytotoxic effects in the tested cell lines. In drug combinations, talazoparib significantly sensitized MDCKII-ABCC1 cells (R_R_ = 2.14) and A431-ABCC1 cells (R_R_ = 1.69) to daunorubicin (Figure 6A,C and Table 1). Talazoparib also behaved as an efficient MDR reversal agent in MDCKII and A431 models overexpressing human ABCG2 transporter (R_R_ = 5.28 and 2.83, respectively) (Figure 6B,D and Table 1). In the parent cells, however, we found no statistically significant shifts in the IC_50_ values of daunorubicin (R_R_ = 0.917 and 0.909 for MDCKII and A431 cells, respectively) or mitoxantrone (R_R_ = 1.06 and 1.00) (Figure 6A–D and Table 1). According to the differences in R_R_ values between transporter-overexpressing and parent cells, transporter inhibition was the most important factor causing the observed resistance modulation. Additionally, Chou and Talalay’s combination index method was used for the exact quantification of drug combination outcomes. The combinations showed synergistic CI values at almost the whole F_A_ range in cells with ABCB1 and ABCG2 overexpression, while antagonism or additivity was detected in parental sublines (Figure 6E–H). This differential pattern confirms the crucial role of cytostatic efflux inhibition in the recorded MDR reversal. At the same time, these results suggest that talazoparib might be a highly effective dual-activity MDR antagonist in cancer cells with ABCC1 and/or ABCG2 overexpression.

### 2.5. Talazoparib Targets Cytostatic MDR in Patient-Derived NSCLC Explants Ex Vivo

Next, we investigated whether observed chemosensitizing potential of talazoparib might have some impact in a clinically relevant model. We chose eight patient-derived NSCLC explants exhibiting differential expressions of ABCG2 and ABCC1 (Figure 7A). Talazoparib and probe inhibitors significantly inhibited cytostatic efflux in primary cultures with high expression of examined transporters, whereas insignificant accumulation changes were recorded in those showing low expression (Figure 7B). Noteworthily, we also found a clear association between the results of drug combination experiments and accumulation studies. Primary cultures with significant drug accumulation changes produced synergistic or additive effects following combined drug treatment. Conversely, antagonistic outcomes were predominantly monitored in the samples exhibiting negative accumulation results (Figure 7C). Taken together, our ex vivo results suggest that talazoparib might serve as an effective dual-activity MDR-reversal agent in patients suffering from tumors with high ABCG2/ABCC1 expressions.

### 2.6. MDR-Related Transporters Do Not Establish Resistance to Talazoparib

Herein, we aimed to investigate possible MDR-victim properties of talazoparib, i.e., whether functional presence of ABC transporters could negatively affect its antiproliferative effects. Comparative MTT studies were conducted in MDCKII and A431 cells with and without transporter overexpressions. As demonstrated in Figure 8A, we observed no significant differences in talazoparib’s anticancer effects between MDCKII-par cells (IC_50_ = 69.4 µM) and the corresponding ABCB1- (IC_50_ = 67.5 µM), ABCC1- (IC_50_ = 66.0 µM), and ABCG2-overexpressing counterparts (IC_50_ = 64.5 µM). An identical pattern was recorded in A431 cells, where parental variant showed sensitivity comparable to that from ABCB1-, ABCC1-, and ABCG2-expressing sublines (IC_50_ of 86.8, 92.7, 91.5, and 91.8 µM, respectively) (Figure 8B). Based on these outcomes, we can presume that the activities of MDR-related ABC transporters are not associated with the establishment of resistance to talazoparib.

### 2.7. Talazoparib Does Not Change the Expressions of MDR-Related Transporters in Breast Cancer Model

Talazoparib might theoretically potentiate a pharmacokinetic MDR profile of target cancer cells via the upregulation of MDR-associated ABC transporters. Thus, in the final experimental set, we monitored the effect of talazoparib on the expressions of *ABCB1*, *ABCC1*, and *ABCG2* in breast cancer MCF-7 cells. The negligibly toxic talazoparib concentration (0.5 µM) was chosen based on the results of an MTT viability experiment (Figure 9A). In induction studies, talazoparib did not trigger the transporters’ expression increase or decrease by more than 100% or 50%, respectively, in comparison with vehicle control (Figure 9B). Therefore, we can conclude that the tested PARPi is not likely to potentiate pharmacokinetic MDR profile in the target breast tumor cells.

## 3. Discussion

DNA repair enzymes PARP1 and PARP2 have emerged as clinically useful targets. PARP inhibitors, including talazoparib (sold as Talzenna), have recently revolutionized the therapy of ovarian and breast cancers [10]. In this study, we investigated pharmacokinetic inhibitory interactions of talazoparib and explored its role in MDR in vitro and ex vivo. 

First, we launched our study with the investigation of talazoparib’s inhibitory affinity toward selected ABC drug efflux transporters and CYP isoenzymes. Based on our results, talazoparib might potentially cause pharmacokinetic DDIs on ABCC1 and ABCG2, but not on ABCB1 or any of the eight tested CYP isoenzymes. To the best of our knowledge, we are the first to provide detailed overview of inhibitory properties of talazoparib toward ADME- and/or MDR-related proteins. According to the in-human mass-balance study, talazoparib is minimally metabolized, and a major fraction of the dose is excreted renally and biliary in unchanged form [30]. Thus, competitive inhibitory interactions on CYPs are impossible to occur, which correlates with our findings.

In the next step, we explored possible exploitation of observed inhibitory interactions for the negative affection of cytostatic MDR. In in vitro drug combination studies, we provided mechanistic evidence on MDR-combating abilities of talazoparib. Importantly, synergistic potentiation of ABCC1- and ABCG2-substrate cytotoxic drugs was also found in primary ex vivo NSCLC explants with clear dependence on the functional expression of the examined transporters. Observed synergistic outcomes have a crucial impact for the practical use of suggested drug combinations; synergism substantially improves the safety of therapy by allowing for drug dose reduction in oncological clinical practice [31]. Taken together, these results suggest that talazoparib might gain a position in combined therapy of ABCC1/G2-overexpressing *BRCA*-mutated tumors as a valuable dual-activity MDR modulator. To verify the rationality of this statement, the performance of xenograft and in-human clinical studies will be necessary. Importantly, both ABCG2 and ABCC1 were described as MDR mediators in breast cancer at THE clinical level. High ABCG2 expression was found in A significant number of breast cancer patients and was correlated with tumor grade, clinical stage, and lymph node metastasis, indicating its association with MDR and possible role as independent prognostic marker [32,33]. Besides ABCG2, ABCC1 is also frequently overexpressed in breast cancer tissues showing correlation with aggressive phenotype and chemotherapeutic resistance [33,34]. So far, talazoparib has been combined with new targeted drugs as well as different conventional chemotherapeutics in clinical trials. Some of these studies focus on anticancer agents that have been recognized as victims of ABCC1/G2-mediated MDR. An upcoming phase I clinical study, NCT05101551, a trial for relapsed pediatric acute myeloid leukemia, aims to determine the safety and efficacy of talazoparib in combination with topotecan. Topotecan is well-known as a sensitive ABCG2 substrate [35]. In addition, it will be interesting to observe the follow-up of the phase I study, which investigated the combination of talazoparib with irinotecan in children and young adults with recurrent/refractory solid tumors [36]. Both ABCC1 and ABCG2 are capable of active efflux of irinotecan and its active metabolite SN-38 [37,38]. Some clinical trials are currently recruiting patients, including NCT03911973, the study evaluating the efficacy of gedatolisib in combination with talazoparib for the therapy of advanced breast cancer. Gedatolisib is a promising anticancer drug candidate that is an MDR victim through the ABCG2 transporter [39]. Apart from clinical trials, some preclinical studies have demonstrated the synergistic activity of talazoparib’s combination with MDR victim drugs. Lok and colleagues demonstrated that talazoparib synergizes with temozolomide in vitro in small cell lung cancer cell lines and showed strong combinatorial efficacy in vivo in patient-derived xenograft models [40]. Temozolomide was characterized as an MDR victim on ABCG2 transporter [41]. In addition, temozolomide, SN-38, and doxorubicin synergized with talazoparib in osteosarcoma cell lines bearing molecular features of *BRCA1/2*-mutated tumors [42]. Based on our results, it might be beneficial to analyze the data from the abovementioned clinical and experimental studies in the context of ABCC1/G2 expressions and their possible associations with studies’ outcomes.

Besides revealing the MDR-modulatory properties in drug combination assays, comparative proliferation studies were conducted to understand its possible opposite feature, i.e., the susceptibility to pharmacokinetic resistance. Our results demonstrate that functional expressions of ABCB1, ABCC1, and ABCG2 are not associated with the emergence of resistance toward talazoparib. Previous brain penetration studies in Abcb1- and Abcg2-knockout mice models and accumulation experiments in MDCKII-ABCB1 and MDCKII-ABCG2 cells showed that talazoparib is a substrate of ABCB1, but not of ABCG2 [43]. The role of ABCB1 in the control of talazoparib’s disposition has been further confirmed in the clinical DDI study with ketoconazole and rifampicin (ABCB1 inhibitor and inducer, respectively) [44]. The discrepancy between the proved ABCB1 substrate affinity and MDR establishment failure has been frequently seen in our recent studies with different novel drugs [45,46,47] as well as in the works by other teams [48,49]. As explained in detail in our papers, several factors can participate in observed discordance, such as relatively high lipophilicity of talazoparib (logP ≈ 2.93 according to ALOGPS 2.1). In addition, in the study with alisertib, we provided the direct evidence that incubation time differences between comparative proliferation assays and transport studies may result into this conflicting situation [47]. However, in the light of our results, talazoparib’s MDR victim role is unlikely, which makes this PARPi a promising MDR modulator.

Finally, we investigated the short-term effects of talazoparib on the mRNA levels of clinically relevant ABC efflux transporters. In our qRT-PCR experiments, talazoparib did not cause significant transcriptional changes in *ABCB1*, *ABCC1*, and *ABCG2* genes in the MCF-7 breast cancer cells. In contrast, long (12-days) exposure to talazoparib has been found to upregulate *ABCB1*, *ABCC1*, and *ABCG2* mRNAs in another breast cancer model, MCF-10A. Interestingly, protein levels have been fluctuating (concentration-dependent induction or downregulation), unchanged, and decreased for ABCB1, ABCC1, and ABCG2, respectively [50]. However, considering our data and the discussed ones, it is not presumable that talazoparib has significant potential to potentiate pharmacokinetic MDR in target tissues. At the same time, downregulation of ABCG2 might be a beneficial factor supporting the talazoparib-mediated MDR reversal. Nevertheless, breast cancer patients are treated with talazoparib for several months [11]; thus, data from clinical DDI studies could provide a definite information on this issue.

## 4. Materials and Methods

### 4.1. Chemicals and Reagents

Talazoparib was obtained from MedChemExpress (New Jersey, NJ, USA). TaqMan systems for detecting *ABCB1*, *ABCG2*, *ABCC1*, *B2M*, and *GAPDH*, and TaqMan™ Universal Master Mix II (no UNG) were obtained from Applied Biosystems Life Technologies (Carlsbad, CA, USA). TRI Reagent for RNA isolation was acquired from the Molecular Research Center (Cincinnati, OH, USA). Oligo (dT) was obtained from Generi Biotech (Hradec Kralove, Czech Republic). Deoxynucleotide (dNTP) Solution Mix and Proto-Script^®^ II Reverse Transcriptase were bought from New England Biolabs (Ipswich, MA, USA). Opti-MEM, Minimum Essential Medium (MEM), media for primary culture (Dulbecco’s Modified Eagle Medium (DMEM): Nutrient Mixture F-12) were bought from Gibco BRL Life Technologies (Rockville, MD, USA). MK571 and Ko143 were received from Enzo Life Sciences (Farmingdale, NY, USA). LY335979 (zosuquidar) was gained from Toronto Research Chemicals (North York, ON, Canada). Anti-β-actin (cat. no. ab8226) and mouse monoclonal anticytokeratin 18 antibody [C-04] (FITC) (cat. no. ab52459) were attained from Abcam (Cambridge, MA, USA). Primary antibodies against human ABCB1 (cat. no. sc-13131), ABCG2 (cat. no. sc-377176), ABCC1 (cat. no. sc-18835), and secondary antimouse antibody (cat. no. sc-516102) were bought from Santa Cruz Biotechnology (Dallas, TX, USA). Pierce™ Coomassie Plus (Bradford, UK) Assay Reagent (cat. no. 23238), and Vivid CYP3A4 Screening Kit were gained from Thermo Fisher Scientific (Waltham, MA, USA). Dimethyl sulfoxide (DMSO), mitoxantrone, daunorubicin, 3-(4,5-dimethyl-2-thiazolyl)-2,5-diphenyl-2H-tetrazolium bromide (MTT) dimethyl sulfoxide (DMSO), phosphate-buffered saline (PBS), fetal bovine serum (FBS), CYP inhibitors (α-naphthoflavone, miconazole, montelukast, sulfaphenazole, quinidine, and ketoconazole), hormones, penicillin/streptomycin, pituitary extract, triiodothyronine, phosphoethanolimine, ethanolamine, growth factors, gentamicin, collagenase, bovine serum albumin, trypsin inhibitor, protease inhibitor cocktail, Ficoll Paque Plus, and cell culture media were purchased from Sigma Aldrich (St. Louis, MO, USA).

### 4.2. Cell Cultures

Two cell line subsets (MDCKII and A431) with and without ABCB1, ABCC1, and ABCG2 overexpression were included in the study. Canine parental MDCKII (MDCKII-par), MDCKII-ABCB1, MDCKII-ABCG2, and MDCKII-ABCC1 were provided by Dr. Alfred Schinkel (The Netherlands Cancer Institute, Amsterdam, Netherlands). Parental A431 cells (derived from human squamous carcinoma) and their A431-ABCB1, A431-ABCC1, and A431-ABCG2 sublines were kindly donated by Dr. Balasz Sarkadi (Hungarian Academy of Sciences, Budapest, Hungary). Human breast adenocarcinoma MCF7 cells were purchased from the European Collection of Cell Cultures (Salisbury, UK). All the in vitro cell lines were cultivated in high-glucose Dulbecco’s modified Eagle’s medium (DMEM) supplemented with 10% fetal bovine serum (FBS) without antibiotics. Cells were used at passage numbers up to 25, and possible mycoplasma infection was routinely tested. The dimethyl sulfoxide (DMSO) solvent was used for the dissolution of talazoparib, while its concentrations in cell media were ≤ 0.5%. The possible interfering effects of DMSO were eliminated by including vehicle controls when appropriate.

### 4.3. Generation of Primary Explants from Patients’ NSCLC Tissue Samples

NSCLC specimens from lung lobectomies were donated by patients at the Department of Cardiac Surgery, University Hospital Hradec Kralove. All the patients signed the informed consent; its formulation was approved by the University Hospital Ethics Committee (study no. 202002 S04P). Subjects’ features are presented in Table 2. The NSCLC malignant tissues were excised from lung lobes by the pathologist immediately after the surgery. Then, NSCLC primary explants were generated according to the established protocols [45,51]. First, the tumor tissues were cut into small pieces by scalpel. Then, the cells were released from the tissue by the 0.1% collagenase in 1 × MEM at 37 °C for 30 min. Subsequently, 1% BSA in MEM (v/v) was added, and cells were filtered through a 40 μm pore-sized cell strainer. The resulting solution was centrifuged at 200 × g for 5 min and the pellet was resuspended in 1% BSA in MEM. The centrifugation (100 × g for 10 min) in Ficoll Paque Plus was used for removing biological impurities such as cell debris, etc. Next, the NSCLC cells were collected and transferred into c-based media. The additional centrifugation (200 × g for 5 min) was applied to remove the traces of Ficoll solution. In the final isolation step, the pellet was resuspended in c-based media, and primary cells were transferred into a collagen I-coated flask. After cell recovery, fibroblasts were eliminated via antifibroblast microbeads (Miltenyi Biotec, Bergisch Gladbach, Germany). Following an additional recovery period, the physiological cells were removed by the introduction of 10% FBS into the c-based media. Explants’ passage number did not exceed 4 during experiments to retain the characteristics of primary tumor cells.

### 4.4. Accumulation Assay with Fluorescent Cytostatic Substrates

The conduction of accumulation assays followed the slightly changed protocol described previously [46,52,53,54]. In the study, MDCKII-par, MDCKII-ABCB1, MDCKII-ABCC1, MDCKII-ABCG2, and primary explants were seeded into 12-well plates in variant densities, 22.0 × 10^4^, 15.0 × 10^4^, 25.0 × 10^4^, 22.0 × 10^4^, and 15.0 × 10^4^ cells/well, respectively. After 24 h incubation, two washing steps with 1 × PBS were applied. Then, different talazoparib dilutions or model inhibitors (LY335979 for ABCB1; MK571 for ABCC1 and Ko143) in Opti-MEM solutions were added to the cells. Accumulations were started by adding 0.5 µM mitoxantrone or 2 µM daunorubicin as the probe substrates of ABCG2 or ABCC1/ABCB1, respectively, after a 10-minute preincubation period. The cells were then incubated under standard conditions for 30 min. Next steps included placing the plates on ice, washing the cells with ice-cold 1 × PBS two-times, and detaching the cells with the ice-cold 10 × trypsin without phenol red. Following trypsinization, cells were resuspended in 1 × PBS containing 2% FBS. Subsequently, cell suspensions were transferred into the test tubes and kept on ice. The fluorescence in cells was measured immediately after the collection of their suspensions using flow cytometer Sony SA3800 Spectral Cell Analyzer (Sony Biotechnology, San Jose, CA, USA). SA3800 software (Sony Biotechnology, San Jose, CA, USA) was used to evaluate the data.

### 4.5. Molecular Docking 

Talazoparib was downloaded from the ZINC Database (http://zinc.docking.org) [55]. Its conformation was optimized using the GlycoBioChemPRODRG2 Server [56], and the energy was minimized with the help of UCSF Chimera 1.9 [57]. ABCG2 (PDB IDs: 6HIJ, 6HBU) and ABCC1 (PDB IDs: 5UJA, 6BHU) were downloaded from the RCSB Protein Data Bank (PDB; http://rcsb.org/pdb/) [28,58,59,60,61]. As described previously, ABCG2 deposited under PDB ID: 6HBU contains the E211Q mutation. Therefore, the Swiss Model Workspace accessible via the Expasy Web Server [62] was used to change Gln-211 to Glu-211 using the primary sequence of human ABCG2 (Q9UNQ0) and the PDB ID: 6HBU as a template [46]. The Swiss-Model Workspace was also used to generate a homology model of the ABCC1 transporter. For this purpose, the primary human ABCC1 sequence (P33527) and the crystal structures of the bovine ABCC1 transporter were used. The transporters and talazoparib were further prepared for docking using MGL Tools 1.5.6 [63]. All the ligands were removed; hydrogens and Gasteiger charges were added. AutoDock Vina 1.1.2 [64] was used for the calculations. In the case of ABCG2, talazoparib was docked into the ligand-binding cavity (PDB ID: 6HIJ; x = 129.81, y = 129.91, and z = 142.89; grid box: 25 × 25 × 25; flexible residues: Phe-432, Phe-439, Leu-539, Ile-543, Val-546, and Met-549) and NBDs (PDB ID: 6HBU; x = 113.32, y = 92.03, and z = 129.89, and x = 94.09, y = 115.37, and z = 129.96; grid box: 25 × 25 × 25; flexible residues: Thr-82, Lys-86, Gln-126, Glu-211, and His-243) as previously described [46]. Eight residues were set as flexible (Lys-332, Tyr-440, Trp-553, Phe-594, Arg-1197, Asn-1245, Trp-1246, and Arg-1249) when talazoparib was docked into the ABCC1 model generated from PDB 5UJA (x = 94.81 y = 59.06, and z = 56.65; grid box: 30 × 30 × 30) [47] and six residues when docked into ABCC1 NBD1 (PDB ID: 6BHU; x = 116.42, y = 136.13, and z = 174.02; grid box: 25 × 25 × 25; flexible residues: Trp-653, Lys-684, Gln-713, Glu-1428, Asn-1429, and Gln-1434) and NBD2 (PDB ID: 6BHU; x = 106.43, y = 159.48, and z = 160.57; grid box: 25 × 25 × 25; flexible residues: Asn-767, Glu-1065, Tyr-1302, Lys-1333, Gln-1375, and Asp-1454). The exhaustiveness parameter was set to 8 for all calculations. Transporter–ligand interactions were evaluated using the Protein–Ligand Interaction Profiler (PLIP) [65] and visualized with PyMOL (The PyMOL Molecular Graphics System, Version 2.5, Schrödinger, LLC).

### 4.6. Incubation Assay for Human Recombinant CYPs 

The procedure for measuring inhibitory activity of talazoparib toward CYP isoenzymes was described previously [46,52,53,54]. Possible interactions of examined PARPi with human CYP3A4, CYP2C8, CYP2D6, CYP1A2, CYP2B6, CYP2C9, CYP2C19, and CYP3A5 were evaluated using commercial Vivid CYP450 Screening Kits. In brief, talazoparib and model inhibitors were diluted in supplied buffer and pipetted into the black 96-well plates. CYP isoenzymes were mixed with the NADPH-regeneration system, and resulting solutions were added to the wells. After 10 min preincubation, the solution of NADP ^+^ and a particular Vivid substrate were added to start the CYP-catalyzed reaction. Fluorescence corresponding to the levels of produced metabolites was detected using the microplate reader Infinite M200 Pro (Tecan, Männedorf, Switzerland) in a kinetic mode (1-minute intervals). The data from the 15 min interval were selected to analyze the results.

### 4.7. MTT Proliferation Assay 

The viability MTT assay was employed in several experimental sets, including induction studies, comparative viability assays, and drug combinations. First, 1.3 × 10^4^ (MDCKII sublines), 1.2 × 10^4^ (A431 sublines), 1.5 × 10^4^ (MCF-7), and 1.0 × 10^4^ (primary NSCLC explants) cells were seeded in 96-well plates. After seeding, cells were left to refresh in incubator for 24 h. Secondly, the medium was removed, and the serial dilutions of the drugs or drug combinations were added to the cells. Next to these variants, a medium with 40% DMSO and vehicle-containing media were tested as nonviable and unaffected viability controls, respectively. After an additional 48 h of incubation, the cells were washed with 1 × PBS and then treated with 1 mg/mL MTT solution in Opti-MEM. Subsequently, cells were incubated for 60 min under standard conditions. After MTT treatment, generated formazan crystals were solubilized with DMSO for 10 min. Wavelengths of 570 nm (formazan) and 690 nm (background) were used for absorbance measurements using a microplate reader Infinite 200 Pro NanoQuant (Tecan, Männedorf, Switzerland).

### 4.8. Drug Combination Assays 

The assays evaluating the effect of combination of talazoparib with conventional MDR-victim cytostatics were conducted in the way which was described previously [45,66]. In vitro cell lines (MDCKII and A431) and primary NSCLC explants were seeded (see densities above) in 96-well plates and incubated for 24 h. After incubation, the medium was removed, and several dilutions of mitoxantrone or daunorubicin with or without 5 µM talazoparib were transferred to the cells (within drug combinations, the drugs were applied concomitantly, not sequentially)**.** Combined cells were incubated with the tested compounds for 48 h, and subsequently, MTT assay was performed to assess cell viability as described above. Besides IC_50_-shift analysis, CompuSyn 3.0.1 software (ComboSyn Inc., Paramus, NJ, USA) was used for the additional analysis of obtained data. This software works with the algorithms of the combination index (CI) method of Chou–Talalay, which is frequently used for the accurate quantification of drug combinations’ outcomes [67]. According to the CI method principle, the combination effects were separated into three categories, including synergistic (CI < 0.9), additive (CI between 0.9–1.1), or antagonistic (CI > 1.1) ones.

### 4.9. Western Blotting Analysis

Western blotting analysis was employed for the assessment of ABC transporters’ expressions in NSCLC explants as described in previous studies [45,66]. First, primary cultures derived from NSCLC tumors (7.5 × 10^5^) were seeded in Petri dishes (dimensions: 60 × 16 mm) and incubated under standard conditions. Once the confluence reached 100%, explants were washed two times with cold 1 × PBS and lysed with cell lysis buffer. Following centrifugation (12000 × g for 30 min at 4 °C), the total protein concentration was determined in pure lysates by the Bradford Assay reagent. Separation of proteins was performed in 8% SDS-PAGE gel. Then, the Trans-Blot Turbo^TM^ Transfer System (Bio-Rad Laboratories, Hercules, CA, USA) was used to transfer them to PVDF membranes. Subsequently, a 5% nonfat dry milk in TBST buffer was used for blocking of membranes at 25 °C for 1.5 h. After the blocking step, membranes were incubated with the specific primary antibodies diluted in TBST buffer (anti-ABCB1 (1:500), anti-ABCC1 (1:500), anti-ABCG2 (1:1000), and anti-β-actin (1:)) overnight at 4 °C. Membranes were inserted into the solution of HRP-conjugated secondary antibody (diluted in TBST buffer at a ratio of 1:2000) and bathed for 1 h at 25 °C. After washing the membranes with TBST buffer 3 times, Immobilon Western Chemiluminescent HRP Substrate (EMD Millipore, Billerica, MA, USA) and Chemi Doc^TM^ MP Imaging System (Bio-Rad Laboratories, Hercules, CA, USA) were used for visualization of the specific proteins’ bands. Finally, ImageJ software (version 1.46r; National Institutes of Health, Bethesda, MD, USA) was used for the quantitative analysis of results.

### 4.10. Gene Induction Studies

Gene expression analyses based on quantitative real-time reverse transcription PCR (qRT-PCR) were performed as described previously [46,52,53]. MCF-7 cell line (3.8×10^5^ cells/well) was seeded in 12-well plates 24 h before induction experiments. The medium was replaced with a fresh medium containing 0.5 µM talazoparib, 25 µM rifampicin, or 0.1% DMSO (vehicle control). After 24 and 48 h intervals, samples were collected by using TRI Reagent. Subsequently, total RNA was isolated from the cells using the chloroform/isopropanol method. Reverse transcription of 1 μg RNA to cDNA was conducted with ProtoScript II Reverse Transcriptase using T100 Thermal Cycler (Bio-Rad Laboratories, Hercules, CA, USA) according to the two-step protocol described previously [66]. The expressions of *ABCB1*, *ABCC1*, and *ABCG2* genes were measured by qRT-PCR using specific TaqMan assays with QuantStudio 6 machine (Life Technologies, Carlsbad, CA, USA). qRT-PCR reactions were run under predefined thermal cycling conditions (95 °C for 10 min, then 40 repeats of a cycle of 95 °C for 15 s and 60 °C for 60 s). *GAPDH* and *B2M* were used as house-keeping genes; the fold-changes in target genes’ expressions were determined according to the 2^-ΔΔCt^ method.

### 4.11. Statistical and Data Analysis

The experimental data were analyzed by using the GraphPad Prism version 8.0.1 (GraphPad Software Inc., La Jolla, CA, USA). One-way ANOVA followed by Dunnett’s post hoc test or the two-tailed unpaired *t*-test were used for calculation of *p*-values (specifications are present in figure legends when appropriate). Differences of *p* < 0.05 were considered statistically significant. Nonlinear regression fitting following sigmoidal Hill kinetics was used for determination of IC_50_ values. All the experiments were independently repeated at least three times, while each of these measurements was accomplished in biological triplicate.

## 5. Conclusions

In conclusion, we described the interactions of talazoparib with ABCC1 and ABCG2 and proved its limited inhibitory properties toward ABCB1 and eight CYP enzymes. In addition, we introduced talazoparib as an effective ABCC1- and ABCG2-targeting dual activity MDR modulator acting both in vitro and ex vivo. Finally, based on our results, talazoparib seems to be a truly prospective MDR antagonist since it lacks MDR victim properties and potentiating effects on MDR phenotype. In addition, the drug might be potentially utilized as a valuable chemosensitizer in patients suffering from NSCLC or other tumors expressing ABCC1 and/or ABCG2. Our in vitro and ex vivo observations could provide a useful foundation for future in vivo studies investigating the combination of talazoparib with MDR victim cytostatics in *BRCA*-mutated and ABCC1/ABCG2-overexpressing tumors.

## Figures and Tables

**Figure 1 ijms-23-14338-f001:**
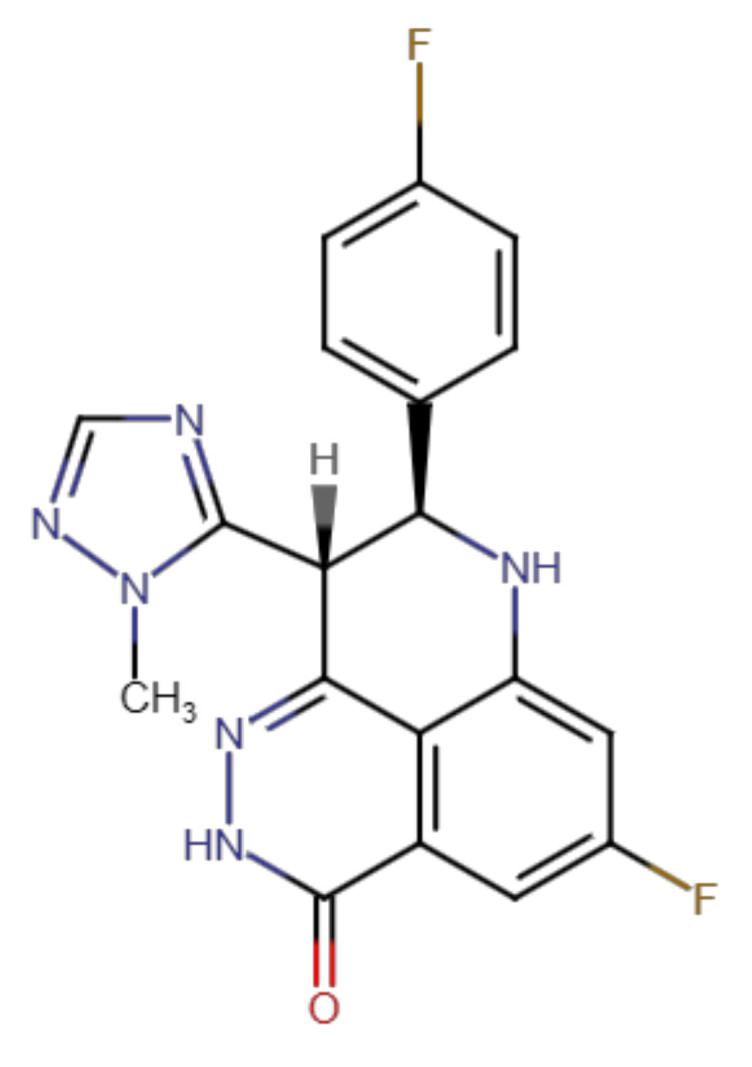
The structure of talazoparib.

**Figure 2 ijms-23-14338-f002:**
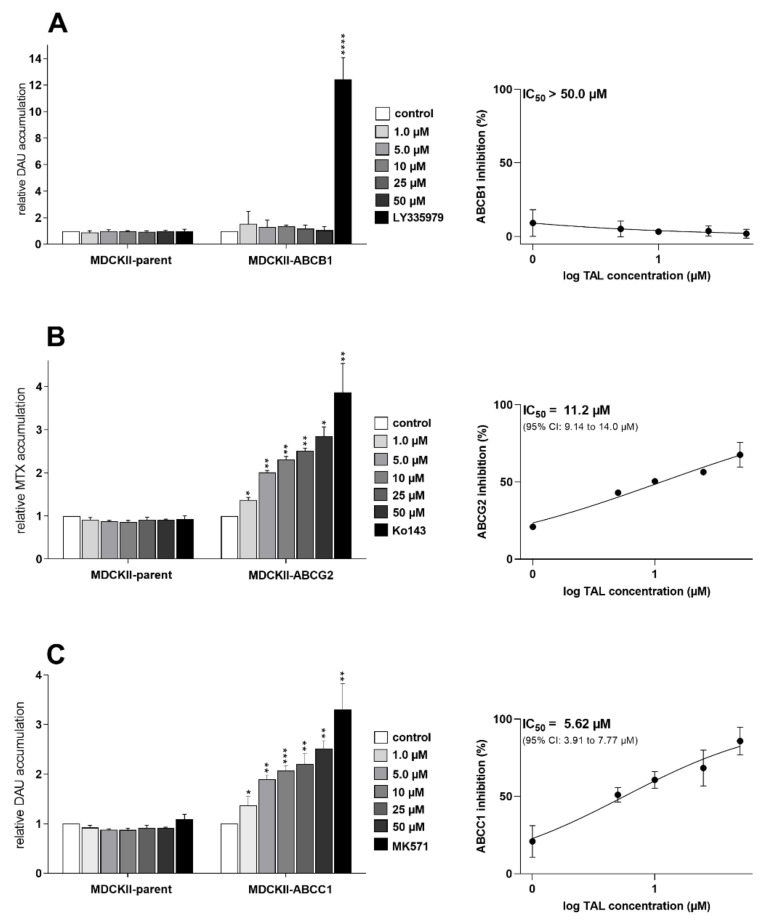
MDCKII cell accumulation studies showing effects of talazoparib on transport activities of (**A**) ABCB1, (**B**) ABCG2, and (**C**) ABCC1. The cells were pre-exposed to talazoparib or model inhibitors (LY335979, Ko143, and MK571, respectively) and then treated with fluorescent chemotherapeutics, daunorubicin, or mitoxantrone. After the substrate accumulation period, the cells were trypsinized, and flow cytometer was used to quantify the intracellular fluorescence. The plotted values of relative accumulation reflect a fold increase in the accumulation of the probe substrate caused by the tested compound; they are numerically expressed as ratios of relative fluorescence units (RFUs) from treated samples to RFUs of vehicle control. Model inhibitors and vehicle controls represented 100% inhibition and 0% inhibition, respectively; their values were applied during normalization of talazoparib data and calculation of its IC_50_ values. Statistical analysis was accomplished using one-way ANOVA followed by Dunnett’s post hoc test (* *p* < 0.05; ** *p* < 0.01; *** *p* < 0.001; **** *p* < 0.0001 relative to control), while the presented data are means ± SD from three independent assays. DAU, daunorubicin; MTX, mitoxantrone; TAL, talazoparib.

**Figure 3 ijms-23-14338-f003:**
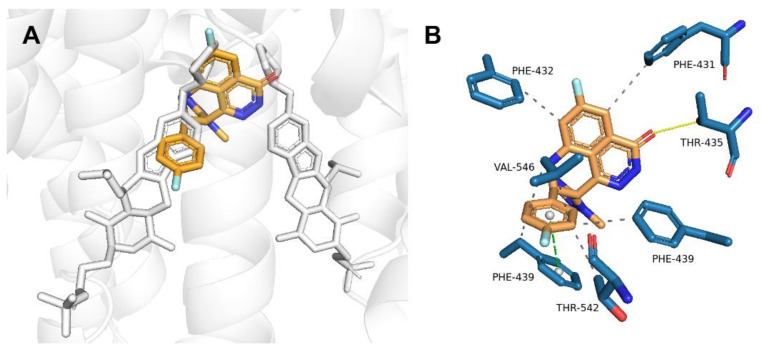
(**A**) ABCG2 with the predicted position of talazoparib (orange) relative to the ligand (gray sticks) that was originally cocrystalized with the transporter (PDB ID: 6HIJ). (**B**) ABCG2 residues (blue sticks) interacting with talazoparib. H-bonds are shown as yellow lines, π-stacking as green-dashed lines, and hydrophobic interactions as gray-dashed lines.

**Figure 4 ijms-23-14338-f004:**
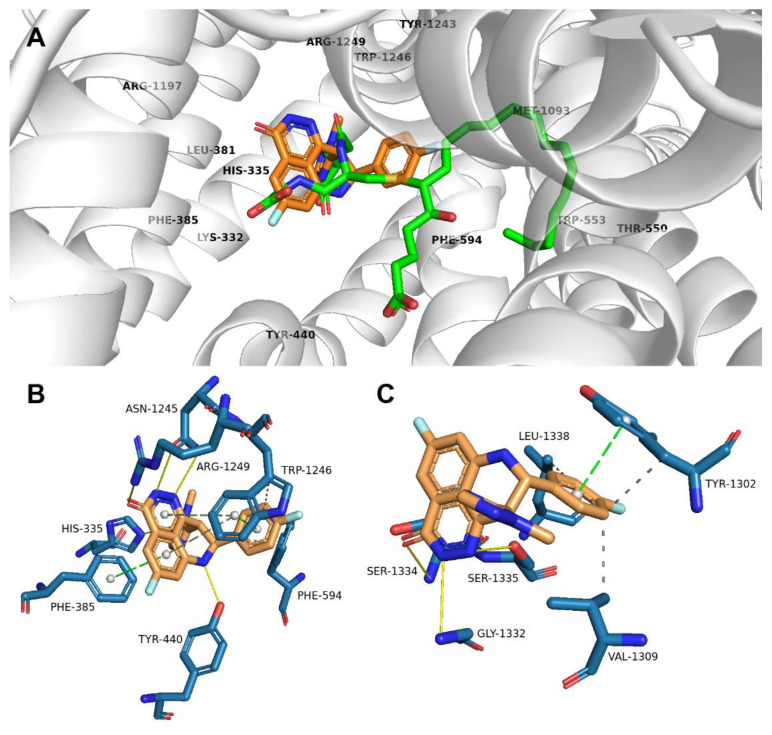
(**A**) Orientation of talazoparib relative to LTC4 (green sticks) in the ABCC1 transporter. (**B**) ABCC1 residues (blue sticks) predicted to interact with talazoparib. (**C**) Predicted interactions of talazoparib with ABCC1′s NBD2. Residues surrounding P- and H-pocket are shown as labels. H-bonds are shown as yellow lines, π-stacking as green-dashed lines, and hydrophobic interactions as gray-dashed lines.

**Figure 5 ijms-23-14338-f005:**
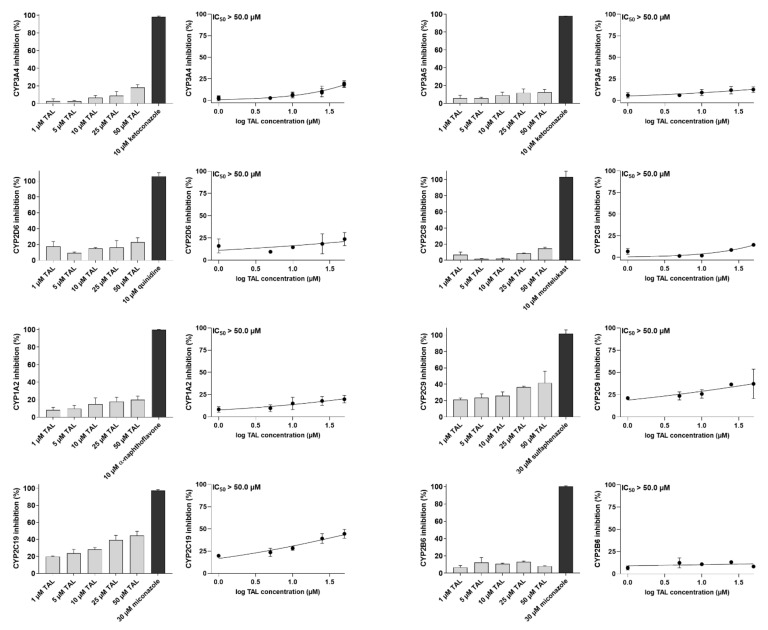
Incubation assays with recombinant CYP1A2, CYP2B6, CYP2C8, CYP2C9, CYP2C19, CYP2D6, CYP3A4, and CYP3A5. The enzymes were pre-exposed to talazoparib or model inhibitors, which was followed by the initiation of reaction using the mixture of NADP ^+^ and particular Vivid substrate. After incubation, fluorescence was measured and then normalized against 0% and 100% activity values. Control 100% activity values were represented by the reaction, which included the enzyme and 0.5% DMSO without tested drug. The control 0% activity values came from the samples that contained only 0.5% DMSO and enzyme solvent buffer without enzyme. IC_50_ values were defined with the normalized fluorescence data. The data come from three independent repetitions and are expressed as means ± SD. TAL, talazoparib.

**Figure 6 ijms-23-14338-f006:**
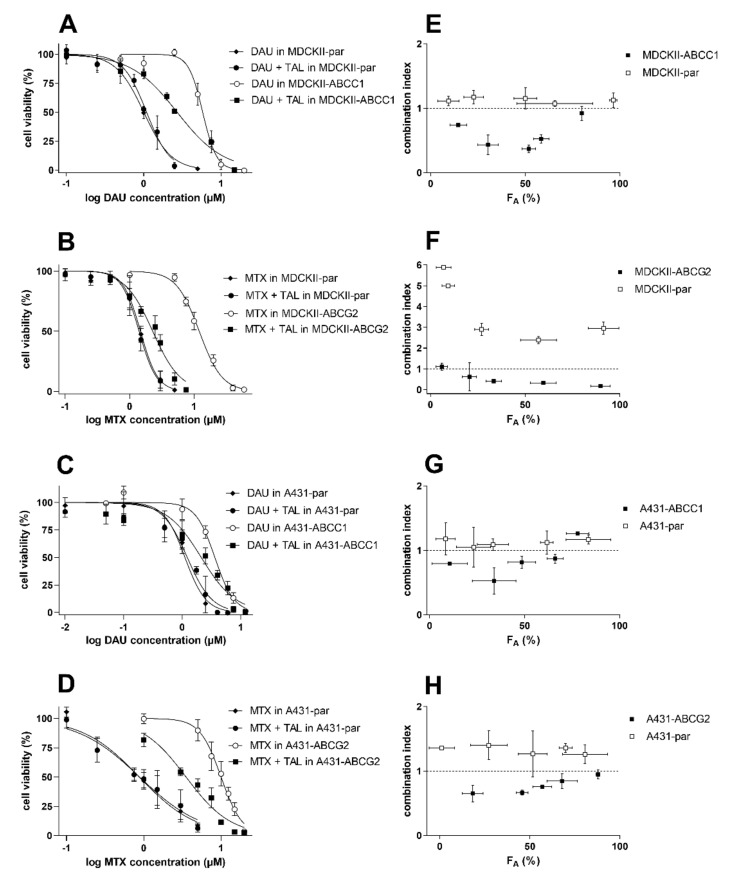
The influence of 5 µM talazoparib on cytotoxic effect of daunorubicin in (**A**) MDCKII-ABCC1 and (**C**) A431-ABCC1, or with mitoxantrone in (**B**) MDCKII-ABCG2 and (**D**) A431-ABCG2 cells. Assays were performed in control parental cells as well. Cells were exposed to drugs alone (daunorubicin, mitoxantrone, or talazoparib) or their combinations and incubated for 48 h. Then, MTT assay was conducted with the cells. Talazoparib concentrations were chosen based on their significant inhibitory effect on the transporters’ activities and acceptable toxicity. CompuSyn software was used to quantify the drug combination outcomes generating F_A_-CI plots for (**E**) MDCKII-ABCC1 and (**F**) MDCKII-ABCG2, and (**G**) A431-ABCC1 and (**H**) A431-ABCG2. Combination effects are synergistic (CI < 0.9), antagonistic (CI > 1.1), or additive (CI between 0.9–1.1). The values were generated in three independent experiments and are plotted as the means ± SD. F_A_, the fraction of cells affected; DAU, daunorubicin; MTX, mitoxantrone; TAL, talazoparib.

**Figure 7 ijms-23-14338-f007:**
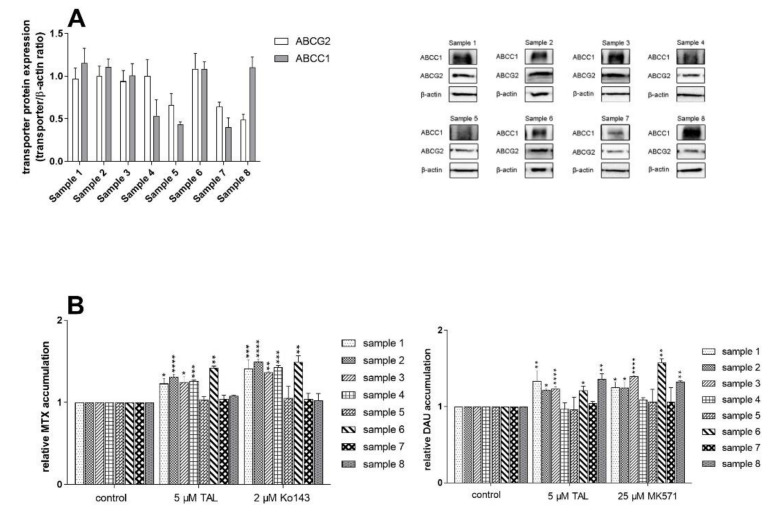
Talazoparib shows MDR-modulatory properties in patient-derived NSCLC primary cultures. (**A**) Western blotting analysis of ABCG2 and ABCC1 expressions (quantitative densitometric analysis on the left, representative pictures on the right). (**B**) Accumulation of mitoxantrone and daunorubicin in the presence of 5 µM talazoparib or probe inhibitors (2 µM Ko143 and 25 µM MK571 for ABCG2 and ABCC1, respectively). One-way ANOVA followed by Dunnett’s post hoc test was used for statistical data analysis (* *p* < 0.05; ** *p* < 0.01; *** *p* < 0.001; **** *p* < 0.0001 compared to control). (**C**) Results of MTT-based combination assays coadministering talazoparib (5 µM) with mitoxantrone or daunorubicin in ex vivo NSCLC explants. Cell viability curves are presented on the left, F_A_-CI plots on the right. CI values below 0.9 express synergism, CIs between 0.9–1.1 additivity, and CIs > 1.1 reflect antagonism. The plotted data were generated in three independent experiments and are expressed as the means ± SD. DAU, daunorubicin; MTX, mitoxantrone; TAL, talazoparib.

**Figure 8 ijms-23-14338-f008:**
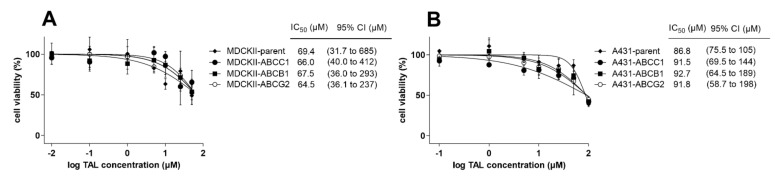
Talazoparib shows comparable antiproliferative effects within (**A**) MDCKII and (**B**) A431 cellular sublines. The cells were exposed to serial dilutions of talazoparib for 48 h, which was followed by viability detection using the MTT proliferation assay. The two-tailed unpaired *t*-test was applied to statistically compare the IC_50_ values from parental cells and their transporter-expressing sublines. However, we did not find any statistically significant differences in either of the cell models. The values in the graphs are means ± SD, which were generated in three independent experiments. TAL, talazoparib.

**Figure 9 ijms-23-14338-f009:**
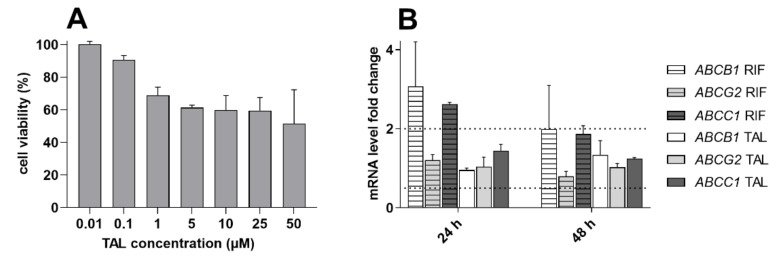
The effect of talazoparib treatment on the *ABCB1*, *ABCC1*, and *ABCG2* mRNA levels in breast cancer MCF-7 model. (**A**) The MTT viability assay performed after the 48 h incubation with a tested drug. (**B**) Gene induction studies. The MCF-7 cells were treated with talazoparib (0.5 µM) or rifampicin (25 µM). The qRT-PCR technique was employed for the assessment of the target genes’ mRNA expressions after 24 and 48 h incubation interval. The dotted lines draw the boundaries for downregulation (lower line) and upregulation (upper line) according to the EMA’s DDI-testing guidelines [24]. The plotted data are means ± SD from three independent repetitions. RIF, rifampicin; TAL, talazoparib.

**Table 1 ijms-23-14338-t001:** IC_50_ values and related IC_50_ shift analysis of the combination of 5 μM talazoparib with cytostatic substrates (daunorubicin and mitoxantrone) in MDCKII and A431 cell lines. IC_50_ values were calculated using the data shown in Figure 6, and possible differences between IC_50_s were subsequently analyzed using the two-tailed unpaired *t*-test (* *p* < 0.05; ** *p* < 0.01; *** *p* < 0.001; comparing IC_50_s from single-drug treatments with IC_50_s of combinations in individual cellular subcultures). The reversal ratio (R_R_) is the ratio of the IC_50_ for single-drug treatment to that for the combined-drug treatment in the particular subline.

Cell Line	Drug(s)	IC_50_ (µM)	95% CI (µM)	R_R_
MDCKII-parent				
	daunorubicin	1.00	(0.878–1.13)	
	mitoxantrone	1.48	(1.35–1.62)	
	daunorubicin + talazoparib	1.09 ^ns^	(1.01–1.18)	0.917
	mitoxantrone + talazoparib	1.40 ^ns^	(1.30–1.53)	1.06
MDCKII-ABCC1				
	daunorubicin	5.80	(5.42–6.19)	
	daunorubicin + talazoparib	2.71 ^**^	(2.36–3.12)	2.14
MDCKII-ABCG2				
	mitoxantrone	12.1	(11.3–12.9)	
	mitoxantrone + talazoparib	2.29 ^***^	(2.04–2.55)	5.28
A431-parent				
	daunorubicin	1.11	(0.966–1.29)	
	mitoxantrone	0.88	(0.714–1.06)	
	daunorubicin + talazoparib	1.22 ^ns^	(1.05–1.43)	0.909
	mitoxantrone + talazoparib	0.88 ^ns^	(0.692–1.11)	1.00
A431-ABCC1				
	daunorubicin	3.62	(3.19–4.10)	
	daunorubicin + talazoparib	2.13 ^*^	(1.70–2.64)	1.69
A431-ABCG2				
	mitoxantrone	10.0	(9.36–10.7)	
	mitoxantrone + talazoparib	3.53 ^**^	(2.99–4.08)	2.83

**Table 2 ijms-23-14338-t002:** Basic specifications of tumor biopsies donated by NSCLC patients.

Sample No.	Gender	Age	Histopathological Diagnosis
1	male	73	adenocarcinoma
2	male	66	squamous carcinoma
3	female	71	adenocarcinoma
4	male	73	squamous carcinoma
5	male	72	adenocarcinoma
6	female	69	adenocarcinoma
7	male	64	combined neuroendocrine carcinoma (large cell + small cell)
8	male	70	squamous carcinoma

## Data Availability

The authors declare that the data generated and analyzed during this study are included in this published article. In addition, datasets generated and/or analyzed during the current study are available from the corresponding author on reasonable request.

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
