# Peer review of "Talazoparib Does Not Interact with ABCB1 Transporter or Cytochrome P450s, but Modulates Multidrug Resistance Mediated by ABCC1 and ABCG2: An in Vitro and Ex Vivo Study"

_ijms, 2022, doi:10.3390/ijms232214338_

Round 1
Reviewer 1 Report
This manuscript focus in the multidrug resistance phenotype o cancer cells, even when promising drugs are used and on the role of ABC transporters in this process, as well as of xenobiotic metabolizing enzymes like Cyp450, in this process. The work investigated the interactions of an antitumor drug, talazoparib, with proteins of these groups. Furthermore, they also examined the role of this PARPi in the MDR phenotype. The subjecto is of great clinical interest and the organization of the paper is good, and in general well written, but some points should be detailed/clarified.
1) In the beginning of the results, line 120, when the authors said "First, we employed MDCKII cell accumulation", they should refer all the cell lines used, before the description of the results of the assays. Also such description should be more clear, mentioning the cell lines throughout the paragraph.
2. In the legend of the figure 1 and 2, the authors should mention how the % of the MDR protein or CYP activity was determined
3. Concerning the results presented in figure 4, the authors treated the cells with Tal+Dau or Tal+Mtx and with Dau or MTX alone. Additional experiments should be done using tal alone
4. Still concerning such results, were the drugs used together or sequentially? Why did the authors choose the concentration of 5uM of Tal for the assay?
5. The auhors refer that probably MDR-related ABC transporters are not associated with the establishment of resistance to talazoparib, presenting results concerning cells expressing ABCB1, ABCC1 and ABCG2. What about other MDR transporters?
Author Response
Please see replies to your comments in attached pdf file.

Author Response

(The authors gave the same response as above.)

Round 2
Reviewer 1 Report
The authors answered all the questions and the manuscript is now suitable to be published
Reviewer 2 Report
The authors have addressed all my comments. I do not have any further concern.